# Robust Principal Component Analysis with Adaptive Neighbors

**Rui Zhang**
Arizona State University
Tempe, AZ, U.S.A.
ruizhang8633@gmail.com

**Hanghang Tong**[*]
University of Illinois at Urbana-Champaign
Urbana, Illinois, U.S.A.
htong@illinois.edu

## Abstract

Suppose certain data points are overly contaminated, then the existing principal component analysis (PCA) methods are frequently incapable of filtering out and eliminating the excessively polluted ones, which potentially lead to the functional degeneration of the corresponding models. To tackle the issue, we propose a general framework namely robust weight learning with adaptive neighbors (RWL-AN), via which adaptive weight vector is automatically obtained with both robustness and sparse neighbors. More significantly, the degree of the sparsity is steerable such that only exact $k$ well-fitting samples with least reconstruction errors are activated during the optimization, while the residual samples, i.e., the extreme noised ones are eliminated for the global robustness. Additionally, the framework is further applied to PCA problem to demonstrate the superiority and effectiveness of the proposed RWL-AN model.

## 1   Introduction

As for the high-quality data reconstruction, principal component analysis (PCA) [16, 4, 7] has been widely investigated. To deal with high dimensional data, conventional PCA methods usually include the data preprocessing, i.e., vectorization of each data point. Nonetheless, the vectorization of the data points could easily incur the curse of dimensionality. Therefore, two-dimensional reconstruction has been brought to the study in the field of image analysis. In sum, equipped with the PCA methods [17, 18, 19], the statistical properties of input data can be retained under the obtained subspace.

In reality, the presence of outliers in data largely reduces the performance of PCA approaches. The existing reconstruction methods usually promote the robustness by exploiting the robust norms as their loss functions [10], e.g., $L_1$-norm and non-squared $F$-norm. More specifically, $L_1$-norm based approaches [5, 14, 9] are developed to alleviate the negative effects of local ill-dimensions. For instance, Li *et al.* [5] proposed the $L_1$-norm based 2DPCA (2DPCA-L1) by optimizing multiple projection directions sequentially. The $L_1$-norm based methods approximate the related optimization problem and therefore often lead to a greedy strategy, which is potentially stuck with heuristic solutions and large computational cost. Luo *et al.* [6, 15] proposed a non-greedy algorithm for an approximate solution to the $L_1$-norm based maximization problem. Moreover, non-squared $F$-norm based methods [10] are developed, where the sum of non-squared $F$-norm reconstruction errors is minimized. Zhang *et al.* [20] optimized the robust non-squared $F$-norm based objective by virtue of a dual problem, where the transitional weight is assigned to each term of the objective.

However, aforementioned robust approaches have lots of limitations. Firstly, all of them depend on different types of loss functions, which are potentially sensitive to outliers. For instance, $L_1$-norm based methods are usually utilized to handle the occluded data with local outliers, while

---

[*]Hanghang Tong is the Corresponding Author

non-squared $F$-norm based approaches are effective to tackle the data with global noises. Secondly, when certain samples are excessively polluted, weak robust methods might be incapable of preventing the degeneration of the reconstruction. Zhang *et al.* [20] addressed this problem by learning a sparse weight via a capped model [8, 2], where the threshold is pre-given to eliminate the terms with larger reconstruction errors. In other words, the performance is sensitive to the choice of threshold. Nonetheless, it is strenuous to search the optimal threshold with frequent inaccuracy. Accordingly, the performance of the existing reconstruction methods is unsatisfactory.

In this paper, we propose a general framework named RWL-AN for learning an adaptive weight vector with both robustness and sparse neighbors. RWL-AN can be further applied to a spectrum of subspace learning approaches via the adaptive-weight strategy. Specifically speaking, RWL-AN assigns a smaller weight to the term with larger reconstruction error automatically to reduce the negative effect of local outliers. Besides the *local robustness*, the weight vector is sparse to prevent the excessive noised terms from degrading the performance of the model. In other words, the degree of the sparsity is steerable such that only specified $k$ samples with least reconstruction errors are effective to eliminate the extreme noised data points for the *global robustness*. By applying the proposed RWL-AN framework to the PCA problem, the superiority and effectiveness of the proposed method are demonstrated both theoretically and empirically.

**Notations:** In the paper, all the matrices are written in uppercase. For matrix $\mathbf{M}$, the $ij$-th element of $\mathbf{M}$ is denoted by $m_{ij}$. The trace of matrix $\mathbf{M}$ is denoted by $Tr(\mathbf{M})$. The $\ell_2$-norm of vector $\mathbf{v}$ is denoted by $\|\mathbf{v}\|_2$. $\mathbf{M}^T$ denotes the transpose operation of $\mathbf{M}$. The Frobenius norm of matrix $\mathbf{M}$ is denoted by $\|\mathbf{M}\|_F$. $\mathbf{M}^\perp$ denotes the orthogonal complement space of $\mathbf{M}$.

## 2 Robust Principal Component Analysis Revisited

Given a dataset $\mathcal{X} = \{\mathbf{x}_1, \mathbf{x}_2, \cdots, \mathbf{x}_N\}$, $\mathbf{x}_i \in \mathbb{R}^d$ represents the $i$-th sample. $\mathbf{X} \in \mathbb{R}^{d \times N}$ denotes the associated matrix of the dataset $\mathcal{X}$. To obtain the optimal mean automatically instead of directly centering the data, Nie *et al.* [10] proposed a robust PCA model from the perspective of low-rank approximation, i.e., minimizing the reconstruction errors with optimal mean as

$$\min_{\mathbf{m}, rank(\mathbf{Z})=k} \sum_{i=1}^N \|\mathbf{x}_i - \mathbf{m} - \mathbf{z}_i\|_2, \tag{1}$$

where variable $\mathbf{m} \in \mathbb{R}^d$ serves as the optimal mean in Eq. (1). $\mathbf{Z} = [\mathbf{z}_1, \ldots, \mathbf{z}_N] \in \mathbb{R}^{d \times N}$ represents the low-rank approximation of $\mathbf{X}$ upon the orthogonal subspace $\mathbf{W} \in \mathbb{R}^{d \times m}$. Via the rank factorization of $\mathbf{z}_i$ on the subspace $\mathbf{W}$, we have $\mathbf{z}_i = \mathbf{W}(\mathbf{v}^i)^T$, where $\mathbf{v}^i \in \mathbb{R}^{1 \times m}$. Therefore, problem (1) can be reformulated into

$$\min_{\mathbf{m}, \mathbf{v}^i, \mathbf{W}^T\mathbf{W}=\mathbf{I}} \sum_{i=1}^N \|\mathbf{x}_i - \mathbf{m} - \mathbf{W}(\mathbf{v}^i)^T\|_2, \tag{2}$$

whose third term within the $\ell_2$-norm is the low-rank reconstructed data. Accordingly, the solution of $\mathbf{v}^i$ could be achieved according to the Karush-Kuhn-Tucker (KKT) condition of problem (4) with respect to (w.r.t.) $\mathbf{v}^i$ as $\dfrac{\partial \sum_{i=1}^N \|\mathbf{x}_i - \mathbf{m} - \mathbf{W}(\mathbf{v}^i)^T\|_2}{\partial \mathbf{v}^i} = 0 \Rightarrow \mathbf{v}^i = (\mathbf{x}_i - \mathbf{m})^T \mathbf{W}.$ Therefore, problem (2) can be addressed by solving the following dual problem:

$$\min_{\mathbf{m}, \mathbf{W}^T\mathbf{W}=\mathbf{I}} \sum_{i=1}^N p_i \|\left(\mathbf{I} - \mathbf{W}\mathbf{W}^T\right)(\mathbf{x}_i - \mathbf{m})\|_2^2. \tag{3}$$

where $p_i \leftarrow \dfrac{1}{2\|(\mathbf{I}-\mathbf{W}\mathbf{W}^T)(\mathbf{x}_i-\mathbf{m})\|_2}$ serves as a transitional weight to be iteratively updated. In other words, the smaller weight would be assigned to the term with larger outliers automatically and vice versa for the robustness.

Motivated by problem (2), Zhang *et al.* extends it to a 2D version to enhance the robustness of 2DPCA. Denote an image dataset $\mathcal{A} = \{\mathbf{A}_1, \mathbf{A}_2, \cdots, \mathbf{A}_N\}$, where $\mathbf{A}_i \in \mathbb{R}^{u_1 \times u_2}$ represents the $i$-th image matrix. Robust 2DPCA method is formulated as

$$\min_{\mathbf{M}, \mathbf{B}_i, \mathbf{U}_1^T\mathbf{U}_1=\mathbf{I}, \mathbf{U}_2^T\mathbf{U}_2=\mathbf{I}} \sum_{i=1}^N \|\mathbf{A}_i - \mathbf{M} - \mathbf{U}_1\mathbf{B}_i\mathbf{U}_2^T\|_F, \tag{4}$$

where $\mathbf{U}_1 \in \mathbb{R}^{u_1 \times d_1}$ and $\mathbf{U}_2 \in \mathbb{R}^{u_2 \times d_2}$ are left and right orthogonal subspaces for dimensionality reduction, respectively. $\mathbf{B}_i \in \mathbb{R}^{d_1 \times d_2}$ denotes a low-dimenional representation of $\mathbf{A}_i$. $\mathbf{M} \in \mathbb{R}^{u_1 \times u_2}$ serves as the optimal mean of input data. Since $\mathbf{B}_i$ is free from any constraint, problem (4) could be rewritten as

$$\min_{\mathbf{M}, \mathbf{U}_1^T \mathbf{U}_1 = \mathbf{I}, \mathbf{U}_2^T \mathbf{U}_2 = \mathbf{I}} \sum_{i=1}^{N} \|\mathbf{A}_i - \mathbf{M} - \mathbf{U}_1 \mathbf{U}_2^T (\mathbf{A}_i - \mathbf{M}) \mathbf{U}_2 \mathbf{U}_2^T\|_F. \tag{5}$$

## 3 Framework of Robust Weight Learning with Adaptive Neighbors

The robust PCA methods mentioned above frequently highlight the robustness and reduce the impact of outliers by developing different metrics, which would possibly lead to various limitations. Although sparsity could also be obtained via the capped model, the performance of the models are often sensitive to the presetting threshold, which is difficult to determine.

In this paper, a framework regarding adaptive weight learning is developed to apply to various reconstruction approaches. The adaptive weight vector can be achieved by the proposed framework with **1) robustness**, i.e., the term with larger reconstruction error is assigned with smaller weight to prevent the outliers from dominating the model; **2) sparsity**, i.e., the images with excessive noises are eliminated to prevent the ill samples from decreasing the performance. Accordingly, the proposed framework for Robust Weight Learning with Adaptive Neighbors (RWL-AN) is formulated as

$$\min_{\mathbf{p} \geq \mathbf{0}, \mathbf{p}^T \mathbf{1} = 1} \sum_{i=1}^{N} p_i g(\mathbf{x}_i) + \gamma p_i^2, \tag{6}$$

where $g(\mathbf{x}_i) \in \mathbb{R}^+$ denotes the reconstruction function under the $i$-th data point $x_i$ with trade-off parameter $\gamma$. $\mathbf{p} = [p_1, p_2, \cdots, p_N]^T$ is the weight vector, where $p_i$ is the weight assigned to the $i$-th reconstruction term. The first term in Eq. (6) indicates that a sample with large reconstruction error should be assigned with a small weight, while the second term is the regularization to avoid trivial solution and over-fitting. It is worth mentioning that an efficient technique is further applied to solving problem (6), such that the weight vector $\mathbf{p}$ has $k$ adaptive neighbors (nonzero entries), i.e., only $k$ best well-fitting samples are activated.

Particularly, the following specific derivation is provided to obtain the closed form solution to problem (6). Denote $g(\mathbf{x}_i)$ by $g_i$, then problem (6) is equivalent to

$$\min_{\mathbf{p} \geq \mathbf{0}, \mathbf{p}^T \mathbf{1} = 1} \sum_{i=1}^{N} \frac{1}{2} \left( p_i + \frac{g_i}{2\gamma} \right)^2. \tag{7}$$

Denote $\mathbf{g} = [g_1, g_2, \cdots, g_N]^T$, then problem (7) can be further rewritten as

$$\min_{\mathbf{p} \geq \mathbf{0}, \mathbf{p}^T \mathbf{1} = 1} \frac{1}{2} \left\| \mathbf{p} + \frac{\mathbf{g}}{2\gamma} \right\|_2^2, \tag{8}$$

where $\mathbf{0} = [0, 0, \cdots, 0]^T \in \mathbb{R}^N$ and $\mathbf{1} = [1, 1, \cdots, 1]^T \in \mathbb{R}^N$. Due to the $\ell_1$-ball constraint $\mathbf{p} \geq \mathbf{0}$ and $\mathbf{p}^T \mathbf{1} = 1$, the Lagrangian function is represented as

$$\mathcal{L}(\mathbf{p}, \lambda, \boldsymbol{\sigma}) = \frac{1}{2} \left\| \mathbf{p} + \frac{\mathbf{g}}{2\gamma} \right\|_2^2 - \lambda(\mathbf{p}^T \mathbf{1} - 1) - \boldsymbol{\sigma}^T \mathbf{p}, \tag{9}$$

where $\lambda \in \mathbb{R}$ and $\boldsymbol{\sigma} \in \mathbb{R}^N \geq \mathbf{0}$ are the Lagrangian multipliers. According to the KKT conditions, the optimal solution to problem (8) satisfies

$$\begin{cases} \frac{\partial \mathcal{L}(\mathbf{p}, \lambda, \boldsymbol{\sigma})}{\partial \mathbf{p}} = 0 \Rightarrow p_i + \frac{g_i}{2\gamma} - \lambda - \sigma_i = 0 \\ \qquad\qquad\qquad\qquad p_i \geq 0 \\ \qquad\qquad\qquad\qquad \sigma_i \geq 0 \\ \qquad\qquad\qquad\qquad p_i \sigma_i = 0 \end{cases}. \tag{10}$$

From the KKT conditions in (10), $p_i, (i = 1, 2, \cdots, N)$ can be summarized as

$$p_i = \left( \lambda - \frac{g_i}{2\gamma} \right)_+, \tag{11}$$

where the operator $(\bullet)_+ = \max(\bullet, 0)$. According to Eq. (11), $p_i$ is non-negative and inversely proportional to $g_i$.

Furthermore, we attempt to determine $\lambda$ and $\gamma$ in Eq. (11). Without loss of generality, we assume $g_1 \le g_2 \le \cdots \le g_N$ and thus have $p_1 \ge p_2 \ge \cdots \ge p_N \ge 0$ based on the negative relationship between $p_i$ and $g_i$ in Eq. (11). When only $k$ neighbors of $\mathbf{p}$ are considered, we have

$$\begin{cases} p_k > 0 \Rightarrow \lambda - \frac{g_k}{2\gamma} > 0 \\ p_{k+1} = 0 \Rightarrow \lambda - \frac{g_{k+1}}{2\gamma} \le 0. \end{cases} \tag{12}$$

By combining Eq. (12) with the constraint $\mathbf{p}^T \mathbf{1} = 1$, we have

$$\sum_{i=1}^{k} \left( \lambda - \frac{g_i}{2\gamma} \right) = 1 \Rightarrow \lambda = \frac{1}{k} + \frac{1}{2\gamma k} \sum_{i=1}^{k} g_i. \tag{13}$$

Based on the constraints in Eq. (12) and result in Eq. (13), the following inequality w.r.t. $\gamma$ can be inferred

$$\begin{cases} \frac{1}{k} > \frac{g_k}{2\gamma} - \frac{1}{2\gamma k} \sum_{i=1}^{k} g_i \\ \frac{1}{k} \le \frac{g_{k+1}}{2\gamma} - \frac{1}{2\gamma k} \sum_{i=1}^{k} g_i. \end{cases} \Rightarrow \frac{k}{2} g_k - \frac{1}{2} \sum_{i=1}^{k} g_i < \gamma \le \frac{k}{2} g_{k+1} - \frac{1}{2} \sum_{i=1}^{k} g_i. \tag{14}$$

To achieve exact $k$ nonzero weights, the upper bound $\gamma = \frac{k}{2} g_{k+1} - \frac{1}{2} \sum_{j=1}^{k} g_j$ is selected. With $\lambda$ and $\gamma$ in Eqs. (13) and (14) respectively, $p_i$ in (11) can be eventually formulated as

$$p_i = \left( \lambda - \frac{g_i}{2\gamma} \right)_+ = \left( \frac{1}{k} + \frac{1}{2\gamma k} \sum_{j=1}^{k} g_j - \frac{g_i}{2\gamma} \right)_+ = \left( \frac{2(\frac{k}{2} g_{k+1} - \frac{1}{2} \sum_{j=1}^{k} g_j) + \sum_{j=1}^{k} g_j - k g_i}{2k(\frac{k}{2} g_{k+1} - \frac{1}{2} \sum_{j=1}^{k} g_j)} \right)_+$$

$$= \left( \frac{g_{k+1} - g_i}{k g_{k+1} - \sum_{j=1}^{k} g_j} \right)_+ . \tag{15}$$

From Eq. (15) regarding the weight $p_i$, we could notice that **1)** $p_i$ is non-negative and inversely proportional to $g_i$, which ensures the *local robustness* of reconstruction problem (6), i.e., the term with larger reconstruction error is assigned with a smaller weight; **2)** if $i > k$, then $p_i = 0$, which ensures the sparsity of $\mathbf{p}$ in problem (6), such that only $k$ terms with smallest reconstruction errors are considered or activated; **3)** $k$ is a steerable integer parameter that directly manipulates the number of activated samples, which indicates a *global robustness* to the outliers. According to Eq. (15), Algorithm 1 is developed by solving the proposed RWL-AN framework in (6).

---

**Algorithm 1:** Algorithm for solving RWL-AN in (6)

---

**Input:** a vector $\mathbf{g} = [g_1, g_2, \cdots, g_N]^T$ that preserves the reconstruction errors under each sample; the integer parameter $k$ ($k \le N$) that controls the number of activated samples.
**Output:** a weight vector $\mathbf{p} = [p_1, p_2, \cdots, p_N]^T$ assigned to each term in the objective (6).

1 Sort $\mathbf{g}$ satisfying $g_1 \le g_2 \cdots \le g_N$;

2 Calculate $p_i = \left( \frac{g_{k+1} - g_i}{k g_{k+1} - \sum\limits_{j=1}^{k} g_j} \right)_+ , (i = 1, 2, \cdots, N)$;

---

# 4 Robust PCA under RWL-AN

Equipped with the RWL-AN framework in (6), we propose the robust PCA model under the proposed RWL-AN as

$$\min_{\mathbf{m},\mathbf{v}^i,\mathbf{p},\mathbf{W}} \sum_{i=1}^{N} p_i\|\mathbf{x}_i - \mathbf{m} - \mathbf{W}(\mathbf{v}^i)^T\|_2^2 + \gamma p_i^2 \tag{16}$$
$$s.t. \ \mathbf{p} \geq \mathbf{0}, \mathbf{p}^T\mathbf{1} = 1, \mathbf{W}^T\mathbf{W} = \mathbf{I},$$

where $\mathbf{W} \in \mathbb{R}^{d \times m}$ is the orthogonal subspaces and $\mathbf{v}^i \in \mathbb{R}^{1 \times m}$ denotes a low-dimensional representation of $\mathbf{x}_i$. Similar as problem (2), the optimal solution $\mathbf{v}^i$ to problem (16) can be derived as $\mathbf{v}^i = (\mathbf{x}_i - \mathbf{m})^T\mathbf{W}$. Specifically speaking, the term $\|\mathbf{x}_i - \mathbf{m} - \mathbf{W}(\mathbf{v}^i)^T\|_2^2$ exactly evaluates the reconstruction error for the $i$-th data point and thus satisfies the definition of $g_i$ in the framework (6). To solve problem (16), we utilize an alternative optimization strategy, i.e., coordinate-block descent method [13].

**Optimize $\mathbf{W}$ & $\mathbf{m}$ by fixing $\mathbf{p}$:** When $\mathbf{p}$ is fixed, problem (16) degenerates to

$$\min_{\mathbf{m},\mathbf{W}^T\mathbf{W}=\mathbf{I}} \sum_{i=1}^{N} p_i\| \left(\mathbf{I} - \mathbf{W}\mathbf{W}^T\right)(\mathbf{x}_i - \mathbf{m})\|_2^2, \tag{17}$$

where $\mathbf{m}$ serves as the mean variable.

**Theorem 1.** *The optimal mean $\mathbf{m}^*$ in problem (17) satisfies the form of*

$$\mathbf{m}^* = \mathbf{X}\mathbf{p} = \sum_{i=1}^{N} p_i\mathbf{x}_i. \tag{18}$$

*Proof.* By taking the derivative of Eq. (17) w.r.t. $\mathbf{m}$ and setting it to zero, we have

$$\left(\mathbf{I} - \mathbf{W}\mathbf{W}^T\right)\left(\mathbf{m}\mathbf{1}^T - \mathbf{X}\right)diag(\mathbf{p})\mathbf{1} = \mathbf{0}.$$

Note that $\left(\mathbf{m}\mathbf{1}^T - \mathbf{X}\right)diag(\mathbf{p})\mathbf{1} = \mathbf{W}\xi + \mathbf{W}^\perp\eta$ via the associated orthogonal decomposition, thus we have

$$\mathbf{W}\xi - \mathbf{W}\xi + \mathbf{W}^\perp\eta - \mathbf{0} = \mathbf{0} \Rightarrow \eta = \mathbf{0}.$$

Due to the constraint $\mathbf{p}^T\mathbf{1} = 1$ and $diag(\mathbf{p})\mathbf{1} = \mathbf{p}$, we could further obtain that

$$\mathbf{m} = \mathbf{X}\mathbf{p} + \mathbf{W}\xi, \tag{19}$$

where $\xi$ is an arbitrary vector. By substituting Eq. (19), problem (17) can be rewritten as

$$\min_{\mathbf{m},\mathbf{W}^T\mathbf{W}=\mathbf{I}} \sum_{i=1}^{N} p_i\| \left(\mathbf{I} - \mathbf{W}\mathbf{W}^T\right)(\mathbf{x}_i - \mathbf{X}\mathbf{p})\|_2^2, \tag{20}$$

which is totally independent of $\xi$. Therefore, we could select $\xi$ as the zero vector for the convenience, such that the optimal mean $\mathbf{m}^*$ is represented as Eq. (18). □

According to Theorem 1, the optimal solution of $\mathbf{m}$ to problem (17) takes the form as derived in (18). Therefore, problem (17) could be further reformulated into

$$\min_{\mathbf{W}^T\mathbf{W}=\mathbf{I}} Tr(diag(\mathbf{p})\left(\mathbf{X} - \mathbf{X}diag(\mathbf{p})\mathbf{1}\mathbf{1}^T\right)^T \left(\mathbf{I} - \mathbf{W}\mathbf{W}^T\right)\left(\mathbf{X} - \mathbf{X}diag(\mathbf{p})\mathbf{1}\mathbf{1}^T\right))$$
$$\Rightarrow \max_{\mathbf{W}^T\mathbf{W}=\mathbf{I}} Tr(\mathbf{W}^T\mathbf{X}\left(\mathbf{I} - diag(\mathbf{p})\mathbf{1}\mathbf{1}^T\right)diag(\mathbf{p})\left(\mathbf{I} - \mathbf{1}\mathbf{1}^Tdiag(\mathbf{p})\right)\mathbf{X}^T\mathbf{W}) \tag{21}$$
$$= \max_{\mathbf{W}^T\mathbf{W}=\mathbf{I}} Tr(\mathbf{W}^T\mathbf{X}\mathbf{D}\mathbf{X}^T\mathbf{W}),$$

where $\mathbf{D} = diag(\mathbf{p}) - \mathbf{p}\mathbf{p}^T$. Hence, $\mathbf{W}$ are the $k$ eigenvector matrix corresponding to the $k$ largest eigenvalues of $\mathbf{X}\mathbf{D}\mathbf{X}^T$[18].

**Optimize $\mathbf{p}$ by fixing $\mathbf{W}$ & $\mathbf{m}$:** Denote $r_i = \| \left(\mathbf{I} - \mathbf{W}\mathbf{W}^T\right)(\mathbf{x}_i - \mathbf{m})\|_2^2$, then problem (16) could be rewritten as

$$\min_{\mathbf{p}\geq\mathbf{0},\mathbf{p}^T\mathbf{1}=1} \sum_{i=1}^{N} p_i r_i + \gamma p_i^2. \tag{22}$$

Same as problem (6), problem (22) can be solved with the closed form solution as represented in Eq. (15), where $g_i, (i = 1, 2, \cdots, N)$ is replaced by $r_i, (i = 1, 2, \cdots, N)$. $k$ is an integer parameter to determine the number of nonzero weights in $\mathbf{p}$. Similarly, the $i$-th weight $p_i$ is inversely proportional to the associated reconstruction error $r_i$ to promote the local robustness. In addition, as for the $i$-th term satisfying $i \geq (k+1)$, the related weight vanishes, such that the excessive outliers, which might potentially sabotage our model can be totally prevented. In other words, the sparsity promotes the global robustness of the reconstruction problem (16). According to Eqs. (18), (21), and (22), an efficient algorithm can be summarized in Algorithm 2 to solve problem (16). Since the coordinate-block descent method is utilized with achieving the closed form solutions w.r.t. $\mathbf{W}, \mathbf{m}$, and $\mathbf{p}$, Algorithm 2 monotonically converges.

---

**Algorithm 2:** Algorithm for solving robust problem (16)

**Input:** an image matrix $\mathbf{X} = [\mathbf{x}_1, \mathbf{x}_2, \cdots, \mathbf{x}_N]$; the number of effective samples $k$.
**Output:** orthogonal subspace $\mathbf{W} \in \mathbb{R}^{d \times m}$.
1  Initialize random $p$ satisfying $\mathbf{p}^T \mathbf{1} = 1$;
2  **while** *not converge* **do**
3  $\quad$ Update $\mathbf{D} \leftarrow diag(\mathbf{p}) - \mathbf{p}\mathbf{p}^T$;
4  $\quad$ Update $\mathbf{W} \leftarrow \underset{\mathbf{W}^T \mathbf{W} = \mathbf{I}}{\arg\max} Tr(\mathbf{W}^T \mathbf{X} \mathbf{D} \mathbf{X}^T \mathbf{W})$;
5  $\quad$ Update $r_i \leftarrow \| \left( \mathbf{I} - \mathbf{W}\mathbf{W}^T \right) (\mathbf{x}_i - \mathbf{X}\mathbf{p}) \|_2^2, (i = 1, 2, \cdots, N)$;
6  $\quad$ Update $\{p_i\}_{i=1}^N$ by Algorithm 1 with inputting $\{r_i\}_{i=1}^N$;
7  **end**

---

## 5  Experiment

Diverse experiments are conducted to evaluate the performance of our method. Firstly, the experimental settings are provided. Moreover, the experimental results on different tasks are recorded.

### 5.1  Experimental Settings

The proposed robust PCA with RWL-AN is compared to the reconstruction methods including conventional PCA (denoted by PCA) [4], robust PCA with optimal mean (denoted by RPCA-OM) [10], generalized low-rank approximations of matrices (denoted by GLRAM) [18], robust 2DPCA with optimal mean (denoted by R2DPCA) [20] and capped robust 2DPCA with optimal mean (denoted by capped R2DPCA) [20]. The integer parameter $k$ of our method is setted as $[0.85N]$ ($N$ is the total number of data points), such that 85% samples are assigned with non-zero weights. As for capped R2DPCA, $\epsilon$ is searched in the grid of $\{10, 20, \cdots, 50\}$ and the best results are recorded.

Four benchmark face image datasets including AT&T [1], UMIST [3], FEI and FERET [12] are utilized in the experiment. Table 1 reports the information for the benchmark datasets. In each dataset, occlusions are placed with random size (over 25% area) on part of images (number of noised samples = total number of samples × *noise rate*). Note that all the experiments are implemented by MATLAB R2015b on Windows 7 PC with 3.20 GHz i5-3470 CPU and 16.0 GB main memory.

Table 1: The information of the benchmark datasets

| Dataset | AT&T | UMIST | FEI | FERET |
|---|---|---|---|---|
| No. of images | 400 | 575 | 2600 | 1400 |
| Size of images | $64 \times 64$ | $64 \times 64$ | $32 \times 32$ | $80 \times 80$ |
| Class | 40 | 20 | 200 | 200 |

All the methods are evaluated on two tasks regarding image reconstruction and clustering. As for the reconstruction task, numerical results are recorded and compared. As for the clustering task, we employ $k$-means as metric. Moreover, we run 50 times with random initialization in each experiment.

Table 2: Reconstruction error comparison. The best is bolded and runner-up is underlined.

|  | *noise rate* | ours | GLRAM | R2DPCA | capped R2DPCA | PCA | RPCA-OM |
|---|---|---|---|---|---|---|---|
| AT&T | raw | **3.56** | 21.87 | 13.44 | 13.44 | 1197.25 | 5.27 |
|  | 0.2 | **6.65** | 59.45 | 26.91 | 21.88 | 1291.10 | 19.19 |
| UMIST | raw | **4.31** | 23.45 | 16.16 | 16.16 | 674.67 | 6.43 |
|  | 0.2 | **8.50** | 59.18 | 28.13 | 24.11 | 720.49 | 26.66 |
| FEI | raw | **1.35** | 21.26 | 4.29 | 4.29 | 533.63 | 1.99 |
|  | 0.2 | **2.19** | 24.81 | 7.45 | 6.52 | 505.60 | 10.30 |
| FERET | raw | **14.70** | 44.89 | 30.90 | 30.90 | 1661.45 | 19.21 |
|  | 0.2 | **23.26** | 99.61 | 51.32 | 33.20 | 1603.78 | 67.75 |

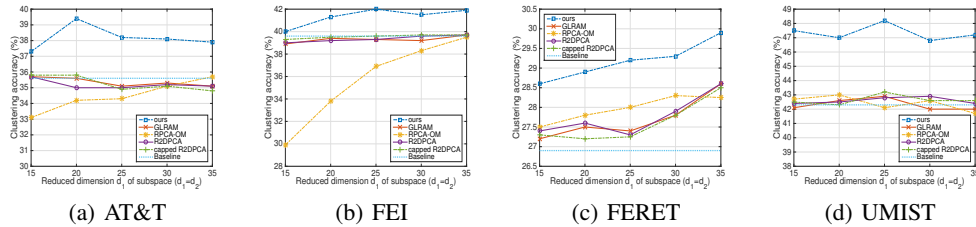

    (a) AT&T           (b) FEI           (c) FERET           (d) UMIST

Figure 1: Clustering accuracy of occluded images and their reconstructed images. The $x$-axis represents the reduced dimensionality $d_1$ of subspace $\mathbf{U}_1$ with the dimensionality $d_2$ of $\mathbf{U}_2$ satisfying $d_2 = d_1$, while $\mathbf{W}$ has the dimensionality $d_1 \times d_2$.

## 5.2 Comparison of Reconstruction Error

Reconstruction problem is to seek the optimal subspace, upon which low-rank images are reconstructed. The performance of the reconstructed methods are measured by $\sum_{i=1}^{N} p_i \|\mathbf{x}_i^r - \mathbf{x}_i^o\|_2^2$ , where $\mathbf{x}_i^r$ represents the $i$-th reconstructed image and $\mathbf{x}_i^o$ is the original image. For the fair comparison, weights are normalized. The reduced dimensionality for 2D method is $d_1 = 9, d_2 = 10$, such that 1D methods perform with the reduced dimensionality $m = 90$. Table 2 records the results of reconstruction error comparison. From Table 2, we could conclude that

1) As for the noised datasets, the proposed method achieves the best performance.

2) As for the raw datasets, RPCA-OM achieves the runner-up performance, while ours and R2DPCA outperform GLRAM and PCA. The results also illustrate the superiority of the optimal-mean based PCA methods.

3) By applying RWL-AN, the reconstruction performance of PCA is largely improved by outperforming all the other competitors. Therefore, the effectiveness of the proposed framework RWL-AN is verified.

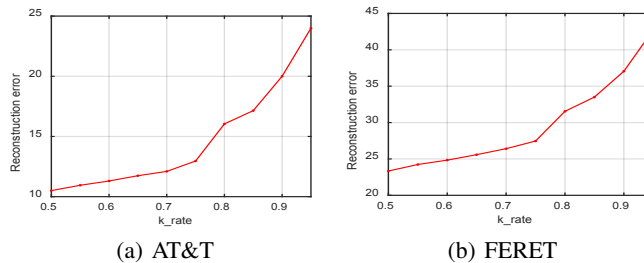

    (a) AT&T           (b) FERET

Figure 2: Reconstruction errors of our proposed method w.r.t the varying parameter $k = N \times k_{rate}$ ($N$ is the total number of samples).

Table 3: CPU Time comparison (seconds) when iteration number is fixed as 50 for each algorithm.

| Method | AT&T | UMIST | FEI | FERET |
|--------|------|-------|-----|-------|
| Ours | 8.14 | 11.40 | 16.76 | 49.03 |
| GLRAM | 6.81 | 8.78 | 14.71 | 45.22 |
| R2DPCA | 8.55 | 12.55 | 17.49 | 49.69 |
| RPCA-OM | 848.46 | 925.73 | 266.58 | 2902.15 |

4) When severe occlusions are involved in the datasets, robust methods as our proposed method, R2DPCA, capped R2DPCA, and RPCA-OM have better performance than the conventional methods including GLRAM and PCA.

From Figure 1, it is noticed that the robust methods as our proposed method, R2DPCA, and capped R2DPCA are superior to GLRAM. The capped R2DPCA overcomes R2DPCA a little, while the proposed method has the outstanding performance under the most cases.

Table 3 reports the CPU time of the comparative algorithms except for capped R2DPCA, which is time-consuming due to tuning an appropriate threshold. We can conclude that the optimal mean based methods including ours, R2DPCA, RPCA-OM are slower than GLRAM due to the calculation of optimal mean in each iteration. Besides that, the time consumption of ours and R2DPCA is similar. In fact, the computation of our weight in Eq. (15) is more complicated than R2DPCA. Nonetheless, due to the sparse weight in the proposed method, ours often runs faster.

### 5.3 Comparison of Clustering

In order to demonstrate the discriminative ability of the reconstructed algorithms, we further compare the clustering results of the reconstructed images via $k$-means classifier, where the clustering accuracy [11] is computed by $ACC = \frac{1}{N}\sum_{i=1}^{N}\delta(l_i, map(c_i))$. $l_i$ denotes real label of the $i$-th instance, and $c_i$ is the corresponding clustering index. $map(\cdot)$ denotes a function that maps each cluster index to the best class label. $\delta(\cdot)$ represents the $\delta$-function, i.e., value is 1 when two input parameters are the same, and 0 otherwise. Figure 1 shows the clustering results under the reconstructed image of different algorithms.

1) Since twenty percent of input images are occluded by noises for each dataset, the superior clustering performance of the proposed method implies its stronger robustness to the outliers.

### 5.4 Sensitivity Analysis w.r.t. Parameter $k$

In this part, the corresponding experiments are conducted to investigate the sensitivity of our model (16) regarding the parameter $k$. We utilize two benchmark datasets known as AT&T and FERET, whose 20% samples are contaminated as previously described. We increase the degree of sparsity by setting $k_{rate}$ from 0.5 to 0.95, where the parameter $k$ is calculated by $N \times k_{rate}$. Moreover, the related reconstruction errors of our proposed method are shown in Figure 2.

1) The curves in Figure 2 are steady when $k_{rate}$ is less than 0.8. Afterwards, the curves increase rapidly, since 20% polluted samples are included.

2) Our model is insensitive to parameter $k_{rate}$, when the $k_{rate} \leq 0.8$, which is the pivotal point. Therefore, we can either determine $k_{rate}$ by tuning it or simply set it as a medium value such as 0.5.

## 6 Conclusion

In this paper, a general framework entitled RWL-AN is proposed, such that the adaptive weight vector is achieved automatically with the local robustness. In particular, the weight vector is sparse with adaptive neighbors, i.e., the degree of the sparsity is steerable with only $k$ activated samples of least reconstruction errors. In other words, the sparsity is steerable to eliminate the excessive noised samples for the global robustness. The framework is further applied to the PCA problem to achieve both local and global robustness. Eventually, theoretical analysis and extensive experimental results are presented to validate the superiority of the proposed method.

**Acknowledgment**

This work is supported by NSF (IIS-1651203, IIS-1715385), and DHS (2017-ST-061-QA0001).

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
