[Reviews · NeurIPS 2019]

Reviewer 1



Update: Thanks for the feedback and I have read them. Yet I don't think it has convinced me to change my decision. For Q2, if the framework is general, the authors should have extended it more than one case. Otherwise, the authors should focus on PCA instead of claiming the framework to be general. For Q3 and Q4, I think the discussion on how to choose k and d is not sufficient in the paper. For Q5, my point is that the naive method (not PCA) will easily beat the proposed method, making the error criterion less convincing. I do like the insights of A1 though. ---------------------- The authors developed a new robust dimensionality reduction framework called RSWL that adaptively learns different weights for different data points to achieve both robustness and sparseness. Here, the sparseness refers to that weights for the data points are sparse (which might be better clarified in the revision). Also, the authors applied RSWL to the PCA problem to develop a robust PCA approach called L3RL. Empirical results are provided to showcase the superiority of the proposed method. I find the idea of learning different weights for different data points interesting and promising. The paper is well-written and the derivations seem to be technically correct. However, I have several concerns listed below: 1. The RSWL framework (7) does not seem to be well justified. Why squared loss for the regularization term instead of l1-norm or l2-norm? Does it work better in practice or it is because the closed-form solution (17) is only available for the squared loss? 2. The authors claim that RSWL is a general framework, yet they only applied it to PCA. How about other forms of the function $f$ other than linear? 3. Can the parameter k be chosen automatically, perhaps via some cross-validation methods? 4. How to choose the reduced dimension d? 5. In line 178, the authors mentioned that the error is quantified via a weighted loss where the weights p_i are learned by the algorithm. Is this fair? Because if it is fair, I imagine a naive method that assigns all weights (i.e. 1) to the data point with the smallest error would easily beat the proposed method. 6. For Figure 1, what is the baseline? 7. If I understand it correctly, the y-value at 0.85 in figure 2a should match corresponding reconstruction error in Table 2 (which is 6.65). Have I missed something?

Reviewer 2



[Review update 08/07] In light of the authors response and discussion with the other reviewers, my opinion about the paper and the score remain the same Originality. The paper is quite interesting and it's main contribution is algorithmic; it provides a very practical algorithm for PCA which tackles outliers. Clarity. The paper is extremely well and clearly written. A couple of minors things I noticed: * Was the the title of the paper meant to say "adaptive weights" instead of "adaptive neighbors"? I don't think the word neighbors is mentioned anywhere in the paper. * The spacing of the Conclusions sections seems to be smaller than 1; please fix for the final version. * Lines 161-163: a very brief description of these other methods and how they differ from the proposed method would be appropriate to include here. Quality. The paper provides sufficient detail on the algorithm and how the update rules were derived, which makes the paper easy to follow and understand. The work is extensively tested using several different metrics and datasets (in fact, this was the most well tested among the papers I reviewed). Significance. The paper addressed a well (and exhaustively) studied area of PCA and low rank approximations. Yet it provides a new perspective for tackling global vs local outliers/noise. As a side observation, it might be interesting to see if the steerable parameter k can be made adaptive as well (i.e. k needs not be the same for each point), which is akin to some of the multi-level/scale PCA methods.

Reviewer 3



Thank you for your feedback. I though the proposed method has an extension over [20] such as a generalization of metrics. However, I don't think the strategy is reasonable. The optimization with two constraints, one of which might be introduced to make the optimization algorithm implementable by well known techniques, causes an underfitting problem. The feedback could not my opinion in my first review. --- This paper considers the problem of robustness against outliers and then developed a new robust PCA. The proposed method optimizes not only low rank matrices but also a mean vector and a weight vector. To avoid the over-fitting problem, the regularization term is also introduced to the cost function. The empirical performance of the proposed method for benchmark datasets looks good. Thus, it would be effective. However, the method would be a moderate extension from related paper [20]. The methodological novelty might be limited.

Reviewer 4



The authors propose a framework for reconstruction tasks (as PCA). The idea is to add weights to all the reconstruction-error (measured point-wise) so when minimizing we will get small weights when the reconstruction-error is large. Also, keep only the top k points whos reconstruction-error is smallest (and hence their weights are the largest). This way the algorithm is less sensitive to outliers, as they will be assigned with small weight and later be chopped out. Then they apply this to the PCA reconstruction function \begin{array} { l } { \min _ { \mathbf { m } , \mathbf { v } _ { i } , \mathbf { p } , \mathbf { w } } \sum _ { i = 1 } ^ { N } p _ { i } \left\| \mathbf { x } _ { i } - \mathbf { m } - \mathbf { W } \left( \mathbf { v } ^ { i } \right) ^ { T } \right\| _ { 2 } ^ { 2 } + \lambda p _ { i } ^ { 2 } } \\ { \text {s.t. } \mathbf { p } \geq \mathbf { 0 } , \mathbf { p } ^ { T } \mathbf { 1 } = 1 , \mathbf { W } ^ { T } \mathbf { W } = \mathbf { I } } \end{array} and then construct an algorithm which optimize for p and for W,m alternately. The algorithm is tested empirically using 4 photo data-sets, where it outperforms all the other robust-PCA methods on all the tests. Although it is concise it seems that the idea is interesting and well treated and analyzed. The main point which bother me is the empirical results. On line 222 - Maybe the pivot point has to do with the polluted ratio being 20%? If so, doesn't that mean that the choice of the noise and the parameter k gave you algorithm a better result then it might have for different k or noise ratio? Besides that I have a few questions: - How is this idea/framework relates to Core-Sets for PCA? - On line 146 - p_i should be r_i? - In the empirical reconstruction error test won’t the plain PCA be the best? Here the whole idea was to ignore noisy points hence accounting them in the measure is problematic. - On line 216 - What do you mean by k_{rate}? Didn't you say (line 164) that k=0.85N?

[Author Response · NeurIPS 2019]

Reviewer # **1**: Q1: The RSWL framework (7) does not seem to be well justified. Why squared loss for the regularization term instead of l1-norm or l2-norm?

**A**1: We would like to point out that if l1-norm or l2-norm is applied to regularization term without square loss, a trivial solution will be obtained due to the linearity of the weight, i.e., linear constraint (l1-ball) and linear index. Although nonlinear entropy term could also be applied to the regularization term, it would not bring the steerable sparsity as the proposed one does to filter out the ill ones.

Q2. The authors claim that RSWL is a general framework, yet they only applied it to PCA.

**A**2: It could also be applied to the graph-based $f = w_{ij}||W^T x_i - W^T x_j||_2^2$ to adaptively construct a sparse Laplacian and fuzzy k-means clustering $f = w_{ij}||x_i - c_j||_2^2$ to learn the sparse fuzzy membership, and etc.

Q3. Can the parameter k be chosen automatically, perhaps via some cross-validation methods?

**A**3: k serves as the key parameter here to filer out the ill ones, i.e., k active and N-k vanishing samples. In this paper, it is manually tuned to different ratio of the noised samples. It would be an interesting future direction to automatically learn hyper-parameter k.

Q4. How to choose the reduced dimension d?

**A**4: Reduced dimension, i.e., low rank is usually chosen between 3%-10% of raw dimension to represent the reconstruction quality, since the reconstruction approximates to the original one when reduced dimension increases.

Q5. In line 178, the authors mentioned that the error is quantified via a weighted loss where the weights $p_i$ are learned by the algorithm. Is this fair? Because if it is fair, I imagine a naive method that assigns all weights (i.e. 1) to the data point with the smallest error would easily beat the proposed method.

**A**5: The comparison is conducted by normalized the weight for all the comparative methods. As for conventional PCA, weight for each term would be 1/N not 1. In other words, the sum of all the weights for the comparative methods is 1 for the fair comparison. Under this setting, smaller error does represent a better reconstruction.

Q6. For Figure 1, what is the baseline? **A**6: Baseline is directly applying k-means to the raw noised data.

Q7. If I understand it correctly, the y-value at 0.85 in figure 2a should match corresponding reconstruction error in Table 2 (which is 6.65). Have I missed something? **A**7: Thanks for the question. Figure 2 and table 2 are under different reduced dimensions.

Additionally, a more detailed literature review of robust PCA is added. We will release the code upon the final version.

Reviewer **#2**: * Was the the title of the paper meant to say "adaptive weights" instead of "adaptive neighbors"?

**A**: Our original meaning is that the adaptive weight vector has only k neighbors to represent the sparsity. We will fix that by developing a reasonable name.

* The spacing of the Conclusions sections seems to be smaller than 1; please fix for the final version.

**A**: Thanks so much for pointing out this. We will fix it as suggested.

* Lines 161-163: a very brief description of these other methods and how they differ from the proposed method would be appropriate to include here. **A**: A brief description with how they differ from the proposed method is added in lines 161-163.

Reviewer **#3**: Thus, it would be effective. However, the method would be a moderate extension from related paper [20].

**A**: We would like to clarify that this paper bears significant difference from reference [20] in the following sense. This paper proposes a general robust framework, whereas reference [20] promotes the robustness by exploiting the robust measure, i.e., capped $l_1$. This paper focuses on the general framework, whose core strategy could be applied to a variety of reconstruction functions to filter out the extremely noised samples. Novelties of methodology are also added to differentiate the proposed method from [20].

Reviewer **#4**: Although it is concise it seems that the idea is interesting and well treated and analyzed. On line 222 - Maybe the pivot point has to do with the polluted ratio being 20%? If so, doesn't that mean that the choice of the noise and the parameter k gave you algorithm a better result then it might have for different k or noise ratio?

**A**: Since k is the activated samples, it only relates to the polluted ratio when all noises seriously pollutes the 20% samples, which should be totally eliminated. However, as for the diverse polluted samples, i.e., some seriously polluted and rest are not, the performance and adaptivity of the proposed method lead to the better results besides the choice of the noise and the parameter k.

- How is this idea/framework relates to Core-Sets for PCA? **A**: The framework could filter out and eliminate the polluted samples, and preserve the well reconstructed ones via adaptive weight, while the conventional PCA is sensitive to the outliers. The framework is specifically proposed to enhance the robustness instead of utilizing the robust measures.

- On line 146 - $p_i$ should be $r_i$? **A**: Thanks for the shrewd observation. We fix it to $r_i$.

- In the empirical reconstruction error test won?t the plain PCA be the best? **A**: Even in unpolluted data, robust PCA still leads to the better reconstruction, since it deals with data reconstruction pointwisely, i.e., assign each data point with different weight to evaluate the importance of each term. Diverse noises are added, such that sensitivity to outliers for each method can be separately evaluated.

- On line 216 - What do you mean by $k_{rate}$? Didn't you say (line 164) that $k = 0.85N$? **A**: $k_{rate}$ is the ratio of choosing value of k from data number N. Varying parameter $k_{rate}$ is shown in Figure 2 to illustrate the impact of different k to the reconstruction.

[Meta-Review · NeurIPS 2019]

The reviews were mixed, but given the competitive nature of the conference, this paper probably doesn't make the threshold. Since the paper deals with adaptive dimensionality reduction, the following paper seems quite relevant: Lee-Ad Gottlieb, Aryeh Kontorovich, Robert Krauthgamer: Adaptive metric dimensionality reduction. Theor. Comput. Sci. 620: 105-118 (2016) Finally, "principle" in the title and abstract should be *principal*.